# Fatty Acids and Bilirubin as Intrinsic Autofluorescence Serum Biomarkers of Drug Action in a Rat Model of Liver Ischemia and Reperfusion

**DOI:** 10.3390/molecules28093818

**Published:** 2023-04-29

**Authors:** Anna C. Croce, Andrea Ferrigno, Giuseppina Palladini, Barbara Mannucci, Mariapia Vairetti, Laura G. Di Pasqua

**Affiliations:** 1Institute of Molecular Genetics, Italian National Research Council (CNR), Via Abbiategrasso 207, 27100 Pavia, Italy; 2Department of Biology & Biotechnology, University of Pavia, Via Ferrata 9, 27100 Pavia, Italy; 3Department of Internal Medicine and Therapeutics, University of Pavia, Via Ferrata 9, 27100 Pavia, Italymariapia.vairetti@unipv.it (M.V.); lauragiuseppin.dipasqua01@universitadipavia.it (L.G.D.P.); 4Internal Medicine, Fondazione, IRCCS Policlinico San Matteo, 27100 Pavia, Italy; 5Centro Grandi Strumenti, University of Pava, 27100 Pavia, Italy; barbara.mannucci@unipv.it

**Keywords:** endogenous fluorophores, arachidonic acid, linoleic acid, bilirubin, liquid optical biopsy, spectral fitting analysis

## Abstract

The autofluorescence of specific fatty acids, retinoids, and bilirubin in crude serum can reflect changes in liver functional engagement in maintaining systemic metabolic homeostasis. The role of these fluorophores as intrinsic biomarkers of pharmacological actions has been investigated here in rats administered with obeticholic acid (OCA), a Farnesoid-X Receptor (FXR) agonist, proven to counteract the increase of serum bilirubin in hepatic ischemia/reperfusion (I/R) injury. Fluorescence spectroscopy has been applied to an assay serum collected from rats submitted to liver I/R (60/60 min ± OCA administration). The I/R group showed changes in the amplitude and profiles of emission spectra excited at 310 or 366 nm, indicating remarkable alterations in the retinoid and fluorescing fatty acid balance, with a particular increase in arachidonic acid. The I/R group also showed an increase in bilirubin AF, detected in the excitation spectra recorded at 570 nm. OCA greatly reversed the effects observed in the I/R group, confirmed by the biochemical analysis of bilirubin and fatty acids. These results are consistent with a relationship between OCA anti-inflammatory effects and the acknowledged roles of fatty acids as precursors of signaling agents mediating damaging responses to harmful stimuli, supporting serum autofluorescence analysis as a possible direct, real-time, cost-effective tool for pharmacological investigations.

## 1. Introduction

The liver carries out various and complex bio-metabolic activities, essential to maintain the blood levels of nutrients and of various signaling biomolecules and providing for catabolic and detoxification functions. These activities ensure systemic metabolic homeostasis, besides counteracting the alterations consequent to the rising of diseased conditions or functional changes, strictly involving several biomolecules able to give rise to autofluorescence (AF) emission under proper excitation light [1]. These endogenous fluorophores can thus act as intrinsic biomarkers of the metabolic and structural properties of cells and tissues, suitable for the development of optically-based analysis procedures with numerous and varied applications [2]. In biomedicine, the outcomes may range from an improved knowledge of metabolic pathway functionality to multipurpose diagnostic applications in the absence of exogenous labeling agents, and the real-time monitoring of their responses to external stimuli and pharmacological treatments [3,4,5].

Numerous endogenous fluorophores can be detected in the liver tissue. Among these, collagen, the fibrous protein of the parenchymal architecture, is overproduced when stellate cells respond to harmful stimuli differentiating to myofibroblasts, making it an inflammation and fibrosis biomarker [6]. Lipofuscin-like lipopigments, composed of the oxidized and crosslinked derivatives of various biomolecules, accumulate as intracellular granules in aging or liver metabolic diseases [7,8,9,10]. The red emission of porphyrins, in turn, can reflect an altered metabolism of heme [11,12,13], while NAD(P)H and flavins, respectively, fluorescing in the reduced and oxidized state, have been the mostly considered AF biomarkers of the redox state of cells for a long time. The strict engagement of these coenzymes of energy metabolism, reductive biosynthesis, and antioxidant activities has made their AF a candidate for both in-vivo monitoring of liver functionality under normal or physiologically altered conditions, or in-situ disease detection [4,14,15,16,17,18,19,20,21,22]. Only some free fatty acids are capable of fluorescing under the near-UV- blue light excitation. Of these, arachidonic acid was firstly assessed as a relevant contributor to the overall AF emission of the liver tissue [23], and then distinguished by means of a spectral fitting analysis from the AF of the miscellanea of different fluorescing lipids in liver lipid extracts. In this case, AF data supported by mass spectrometry results indicated relatively lower levels of arachidonic acid in a rat fatty liver model induced by methyl-choline deficient (MCD) diet administration as compared to a genetic model, which is in agreement with the role of the basal liver metabolic engagement in influencing its lipid composition [24]. 

The subsequent use of variable AF excitation and emission to characterize the fluorescing fatty acids in the solution indicated that the combination of the 310 nm and 366 nm excitations is suitable for discriminating arachidonic, linoleic and oleic acids, retinoids, and proteins and flavins. The translation of the AF detection of fluorescing fatty acids directly to the crude serum allowed for the estimation of their relative changes in a rat model of liver ischemia and reperfusion [25]. In the blood, apart from the attention to the increase in the presence of porphyrin AF as a support in the diagnosis of altered iron metabolism, porphyria, and early cancer [26,27], the AF of fluorescing free fatty acids in the crude serum is promising to meet the increasing need of biomarker serum panels, by improving noninvasive, repeatable procedures for the diagnosis and monitoring of liver diseases and systemic metabolic disorders [28,29,30]. The liver plays an essential role in the storage and mobilization of lipids and retinoids to ensure their homeostasis for the correct fulfillment of several functions in the entire organism, including its important engagement in the regulation of some specific fatty acids as precursors of signaling agents along the pathways modulating the responses of various organs to harmful stimuli [31,32,33,34,35,36,37,38,39,40,41]. Bilirubin has also been proposed as an endogenous fluorophore of blood and bile [42,43,44,45,46]. This takes advantage of the peculiar ability of bilirubin to act as a bi-chromophore, making its AF properties able to be influenced by its interaction with albumin, or with solubilizing agents, such as biliary acids [47,48,49,50]. The chemical structure of bilirubin consists of two unsymmetrical, covalently linked di-pyrrole moieties, bound by a methylene bridge covalent link. This accounts for exciton coupling phenomena and the consequent intramolecular, interchromophore energy transfer, the efficiency of which is influenced by both the microenvironment and the excitation wavelength. Notably, the bilirubin AF excitation profile can be easily discriminated from that of the other serum endogenous fluorophores because of the different spectral positions. In addition, the possibility of solving the bilirubin excitation spectra into different bands allows for an investigation into the changes in their balance, and thus in the related changes in the molecular environment [49,51,52].

Spectroscopy of the AF of crude serum from a rat model of liver ischemia and reperfusion has already indicated the analysis of bilirubin emission and excitation profiles as a valuable, direct parameter of changes in the composition of biological fluids, in the absence of biochemical assay [25,52]. This finding offers basic and interesting insights regarding further investigations into the development of the real-time, cost-effective serum AF analysis as a diagnostic support in assessing the effects of pharmacological treatments. 

In this context, obeticholic acid (OCA), a bile acid-derived semisynthetic Farnesoid X Receptor (FXR) agonist [53] also shown to counteract the increase of serum bilirubin and hepatic matrix metalloproteinases activity in a rat model of hepatic ischemia/reperfusion injury [54,55], has been considered a suitable pharmacological agent to further investigate the changes in serum endogenous fluorophores relatable to liver functional engagement in the maintenance of systemic metabolic homeostasis. 

Therefore, the aim of the present work is to take advantage of the combined AF analysis of fluorescing fatty acids and bilirubin performed in the serum to estimate their changes as reference parameters of the OCA-induced pharmacological effect in a rat model of liver ischemia and reperfusion, as summarized in Figure 1. 

## 2. Results 

### 2.1. Fluorescing Fatty Acids in Serum 

The AF emission spectra collected from all the serum samples under 310 nm excitation consisted of bands covering the 415–590 nm emission, with the maximum peak positions being around 445–450 nm (Figure 1A). A similar spectral range was also covered by the spectra recorded under 366 nm excitation, with the emission maximum position still at around 450 nm (Figure 1B). 

In general, the spectra showed a maximum peak position of around 450 nm with a shoulder in the 490–510 nm region, less marked under excitation at 366 nm than at 310 nm, which allows for an appreciation of its decrease after ischemia and reperfusion compared to the sham-operated rats, as well as its recovery following OCA administration. The relative AF contributions of the different endogenous fluorophores accounting for both the overall emission and the changes in the profile of the AF spectra were therefore estimated by means of the curve-fitting analysis procedure. 

In keeping with previous observations on the specific response of the endogenous fluorophore emission to excitation wavelength, retinol, arachidonic acid, and oleic acid were reliably detected by both the 310 nm and 366 nm excitations, while the spectral component ascribable to linoleic acid was identified under the 366 nm excitation (Figure 2B). The 310 nm excitation also allows for the identification of a spectral component ascribable to the long excitation and long emission tails of proteins. These are mostly represented by albumin, the levels of which have been estimated under its most typical excitation/emission conditions [56], as described later. 

In any case, both the excitations at 310 nm and 366 nm show that the AF emission values relatable to retinol and arachidonic acids, representing the main spectral contributions for every kind of liver treatment condition, show a decrease and an increase following ischemia and reperfusion, respectively. This opposite behavior is reversed by OCA treatment (Figure 2A). 

As to the real values of the measured spectra, the amplitude of the AF emission recorded under 310 nm was, in general, much higher in amplitude than that recorded under the 366 nm excitation. For each kind of excitation, in turn, the serum samples from ischemia reperfusion exhibited the highest values, an effect reversed by OCA administration, resulting in AF amplitude values even lower than those found regarding the sham-operated rats (Table 1). 

These AF absolute data were then used to correct the relative AF values estimated by curve-fitting analysis for the single fluorophores, comparing the real changes in the contribution to the overall emission area of the various fluorophores between the sham-operated and I/R samples in the presence or absence of OCA administration (Figure 3A,B).

Both the 310 nm and 366 nm excitation conditions allowed for the appreciation of an increase in the absolute AF values relatable to retinol at 60 min of ischemia, followed by 60 min reperfusion, and also appreciate a very significant increase in the AF signals relatable to the single fluorescing fatty acids. This effect of I/R was reversed by the treatment with OCA, leading to slightly lower AF values for the fluorescing fatty acids compared to the sham-operated rats. 

The serum levels of the fluorescing fatty acids were also evaluated by gas chromatography-mass spectrometry (GC-MS) analysis (Figure 4).

The levels of the three fatty acids were higher in the I/R group than in the sham-operated group, in keeping with data from AF-fitting analysis. The level of arachidonic acid, in particular, was significantly greater in the I/R group than in the sham-operated group. The administration of OCA, in turn, reduced the serum levels of both arachidonic acid and linoleic acid compared with I/R and sham-operated groups. The same trend, although not significant, occurred for oleic acid (Figure 4). 

### 2.2. Bilirubin Fluorescence in the Serum 

The serum AF emission spectra shown in the previous section (Figure 1A,B) are dominated by the contributions of retinol and fluorescing fatty acids, likely over imposing the emission position of bilirubin at the longer wavelength side of the spectrum. Only in the case of the I/R serum samples excitated at 366 nm did the curve-fitting analysis indicate the presence of a minor band peaking at about 560 nm, contributing for less than 1% of the overall emission area and possibly ascribable to flavins or bilirubin [43,49,57,58]. Conversely, the bilirubin AF profile was more easily discriminated from the other major fluorescing components in the serum excitation spectra, in particular, when spectra are collected at 570 nm [52].

The serum excitation profiles consisted, in general, of a tail decreasing from the shorter wavelength to about 370 nm, followed by a wide band in the 420–520 nm spectral range, and a minor shoulder in the 380–400 nm region (Figure 5A). 

The relative contribution of the bilirubin fluorescence signal to the overall area of the excitation spectrum was estimated by means of the curve-fitting analysis procedure (Figure 5B). 

Biochemical assays showed that the bilirubin total values increased significantly after ischemia /reperfusion in comparison with the sham-operated rats, while a lesser increase occurred in OCA-treated rats, greatly depending on changes in direct bilirubin (Figure 6A). Comparable results were obtained for the AF values relatable to bilirubin (Figure 5B). The increase in the bilirubin AF values after ischemia/reperfusion was also accompanied by a lowering of the (Ʃ > 407 nm)/(404–407) nm ratio values (Figure 6C), as compared with the sham-operated and OCA treated rats. The differences in the ratio values, although not statistically significant, indicated variations in the excitation spectrum profile, likely depending on changes in the microenvironment of the bilirubin molecule. Since the AF emission efficiency of bilirubin is commonly known to be affected by binding with albumin, the levels of this protein in the serum have also been estimated (Figure 6D), taking advantage of its typical AF properties in terms of the maximum position of the emission spectrum at about 340 ± 10 nm under suitable excitation at 280 nm.

## 3. Discussion

In general, the fluorescence-based assays of biological substrates do not provide the real amounts of the biomolecules of interest, contrary to data obtained via many biochemical methods. This limitation, however, does not hinder the value of the fluorescence analysis in providing direct and real-time results when comparative investigations are performed to assess metabolic changes and the alteration of the balance between fluorescing biomolecules, occurring under the development of a disease induced by physiological alterations or pharmacological treatments. In fact, many studies including the AF spectroscopy of crude serum have offered promising perspectives for the development of real-time, non-invasive estimations of fluorescing biomolecules with bio-metabolic functional meanings [25,26,52]. 

The spectral profiles of the serum AF have shown changes depending on the induction of liver ischemia and reperfusion and OCA administration. As has been already assessed, the overall signal of the AF emission spectra excited at 310 nm and 366 nm is mostly ascribable to the contribution of fluorescing fatty acids, namely arachidonic, oleic, and linoleic acids, alongside retinoids and proteins [25]. 

In general, the emission spectra recorded under the 310 nm excitation resulted in a higher emission amplitude than that of the 366 nm excitation. In addition, both excitations showed a significant increase in the signal following ischemia and reperfusion, an effect reversed by OCA treatment. The higher efficiency of the 310 nm excitation in exciting the serum AF was also accompanied by the greatest ability to reveal the differences between the spectral profiles of the different sample groups. This result is accounted for by the differences in the chemical structures of retinol and fluorescing free fatty acids, influencing their spectral positions under the different excitation wavelengths [25,59]. As a consequence, both the choice of the excitation wavelength and the different levels of the single endogenous fluorophores present in the serum influence their contribution to the overall emission spectrum. 

The curve-fitting analysis procedure can discriminate the partially superimposed bands of the single endogenous fluorophores combined in the measured spectra, additionally allowing for an estimation of their relative contribution to the overall emission area. The fitting analysis of the crude serum AF emission spectra indicated retinol and arachidonic acid as being primarily responsible for the overall spectrum recorded under the 310 nm excitation. The relative contribution of retinol was found to decrease in I/R samples compared with the sham-operated ones, compensated by the relative increase in the AF signal relatable to arachidonic acid. The treatment with OCA counteracted this effect. Comparable results were obtained from the fitting analysis of the spectra recorded under the 366 nm excitation, which additionally allowed for the estimation of linoleic and oleic acid contributions, showing their relative increase in I/R in comparison with the samples from the sham-operated rats. These results are thus revealing changes in the balance between the fluorescing fatty acids and retinol, with opposite behavior shown between the relative AF signals ascribable to retinol and arachidonic acid, respectively, decreasing or increasing following ischemia and reperfusion. This effect is partially reversed by the treatment with OCA. 

Additional information is obtained when the percentage values of the relative contribution of each endogenous fluorophore are used to calculate the real AF levels from the respective overall measured AF values. Given the remarkable increase in the overall AF recorded from the serum of rats submitted to liver ischemia reperfusion, the increase in the real contribution of fluorescing fatty acids in this group of samples results in them being even larger than what is shown by percentage data. Also, retinol shows a slight increase with respect to the sham-operated rats. Again, the treatment with OCA is partially reversing these results. The combined evidence given by AF, in terms of the changes in both the levels of fluorescing fatty acids and retinol, as well as in the balance between them, provides a diagnostic means to investigate the activity of the liver in mobilizing these biomolecules to the blood, as well as the bio-metabolic alterations of the entire organism following an injury or pharmacological treatment. 

The AF data on the fluorescing fatty acids, supported by the results of GC/MS analysis, are particularly relevant with respect to the essential role played by the liver in the regulation of arachidonic linoleic acids, and their eicosanoid oxygenated products, including prostanoids, leukotrienes, lipoxins, and hydroxyeicosatetraenoic (n-HETE series), acting as paracrine and autocrine-signaling agents along the pathways of the damaging or protective responses to harmful stimuli [31,32,33,34,35,36,37,38]. Equally important is the role of the liver in the storage and release of vitamin A, or more generally, retinoids, to provide for their availability in the whole organism. The consequent various positive or negative implications account for the efforts devoted to the set-up of a procedure designed to estimate the levels of retinoids in the blood, including the use of cost-effective and fast fluorometric methods [41,60,61,62,63]. 

Bilirubin levels in the blood are deserving of renewed attention. In addition to its common consideration as an additional parameter of liver functionality and the assessed hazardous effects of its high levels in newborns, such as jaundice and related risks of brain damage, increasing evidence of the hormone-like roles of low levels of bilirubin is suggested due to its negative association with obesity and diabetes [64]. Aiming to clarify the reasons for these effects, recent investigations reported on bilirubin’s ability to influence the regulation of peroxisome proliferator-activated receptors and white adipose tissue metabolism, favoring the pathways of lipid-burning and eventually affecting body weight [65]. These investigations could be supported by the detection of bilirubin AF in the serum, which is more facilitated in excitation compared with the emission spectra. In fact, the excitation profile of bilirubin is significantly separate from that of the other fluorophores detectable at shorter wavelengths, as it can be easily observed at 570 nm, the wavelength corresponding to the maximum position of the longer bilirubin emission band [52]. 

The serum AF data indicated a significant increase in bilirubin following ischemia and reperfusion, consistent with the biochemical data and, in particular, with the levels of direct or conjugated bilirubin. This finding is in apparent contradiction with the commonly known enhancement of the bilirubin AF signal upon the binding of its unconjugated form to albumin [66], and can be explained by the decrease of proteins in the ischemia and reperfusion group, as indicated by the AF values estimated at 340 ± 10 nm in the spectra excited at 280 nm, the conditions typical of albumin (Figure 6D). Notably, the loss of albumin is consistent with the literature on the decrease of serum albumin consequent to the experimental induction of liver ischemia and reperfusion associated with albuminuria [67,68,69]. The decreased availability of albumin could explain the slight decrease in the ratio values calculated between the bands at the longer and shorter wavelength positions in the bilirubin excitation spectra recorded from the I/R rat groups. In fact, the albumin binding site for bilirubin is known to result in a molecular environment influencing the efficiency of the bilirubin intramolecular, interchromophore exciton event and the consequent fluorescence signal properties of the bi-chromophore [49,70]. 

## 4. Materials and Methods

### 4.1. Chemicals

The bile acid derivative obeticholic acid (OCA) was kindly provided by Intercept Pharmaceuticals (San Diego, CA, USA).

### 4.2. Animal Model

Male Wistar rats (220–250 g, *n* = 15), Harlan-Nossan, Correzzana, Italy) were maintained at a controlled temperature (21 °C), kept under dark/light cycles of 12 h, and had free access to water and food. The bile acid derivative OCA was administered orally (10 mg/kg/day) in methylcellulose 1% vehicle for 5 days (*n* = 5). 

The liver ischemia and ensuing reperfusion were induced in rats anesthetized with pentobarbital administration (40 mg/kg). Soon after, the abdomen was opened by a median incision and ischemia of liver left and median lobes was induced by clamping the portal vein and hepatic artery with microvascular clips. Sixty minutes later the microvascular clips were removed to allow liver blood reperfusion for 60 min [71,72]. To prevent heat loss during these procedures, the rats were placed on a warm support stand. The control, sham-operated rats (*n* = 5) were submitted to the same anesthesia and liver exposure procedures, but without inducing ischemia. Soon before this, sacrifice blood was collected from the abdominal aorta and immediately centrifuged to separate serum. The serum was immediately frozen in liquid N_2_ and stored at −80 °C until being submitted to AF spectroscopic and biochemical analyses. 

The experimental use and care of the animals were approved by the Italian Ministry of Health and by the University of Pavia Animal Care Commission (Document: 2/2012). To comply with Institutional and European recommendations on obtaining the maximum amount of information from the minimum number of animals, and obtaining the maximum amount of information from each animal, sera were obtained from a rat model used for different experiments.

### 4.3. AF Spectrofluorometric Analysis

The AF spectra were recorded by means of a Spectrofluorometer (LS 55 Model; PerkinElmer Italia, Milan, Italy). On the basis of previous works, the AF ascribable to retinol and fluorescing fatty acids was investigated using the serum AF emission spectra (390–600 nm range) recorded under excitations at 310 nm and 366 nm [25]. The AF ascribable to bilirubin was investigated in the serum AF excitation spectra (300–540 nm) recorded at 570 nm [52]. Emission spectra were also collected in the 390–450 nm interval by exciting at 280 nm, the typical conditions for the selective detection of albumin [56].

For each AF measurement condition, 10 spectra were recorded from each serum sample. 

#### 4.3.1. AF Spectral Fitting Procedure 

The AF spectra recorded to estimate retinol and fluorescing fatty acids, or to estimate bilirubin, were submitted to the curve-fitting analysis procedure to evaluate the fraction of the AF signal of each fluorophore expected to contribute to the overall emission or excitation area. In general, the procedure was based on specific Half-Gaussian Modified Gaussian (GMG) functions, according to the spectral parameters previously defined from the reference emission profiles measured from the endogenous fluorophores [25]. The goodness of the fitting has been verified by means of the analysis of residuals and coefficients of determination. 

Independently from the AF spectra analyzed, the curve fitting analysis procedure was performed using the PeakFit calculation program (SPSS Science, Chicago, IL) based on the Marquardt-Levenberg algorithm [73]; the GMG functions previously defined to describe the endogenous fluorophores expected to be present in the fluid analyzed and contributed to the overall measured emission area. The achievement of a satisfying goodness of fitting was first searched for the AF spectra from sham operated rats, to defined the most satisfying combination of the GMGs; this was then used as a starting condition to proceed to analyze the possible changes in the AF spectra from the I/R and I/R OCA samples.

Before the fitting analysis, the maximum peak values of all spectra were normalized to 100 a.u., ensuring that the contribution of every single spectral component would be expressed as a percentage of the overall area. When this was the case, the percentage values were used to calculate the real, absolute data from the measured, original amplitude values. 

The goodness of fitting was obtained depending on the achievement of the true absolute minimum value of the sum of the squared deviations, and verified in terms of the residual analysis and coefficient of determination (r^2^). 

#### 4.3.2. Serum AF Spectrofluorometric Analysis for Fatty Acid Estimation

The AF analysis of the serum emission spectra has been based on the GMG functions describing the AF emission spectral profiles of retinol and arachidonic, oleic, and linoleic acids, previously defined from the fitting analysis of the AF emission spectra recorded under the 310 nm and 366 nm excitations from the ethanol solutions of the respective pure compounds. These GMG functions were characterized by a specific center peak wavelength position (λ) and their full width at the half intensity maximum (FWHM). 

Briefly, the bands for the emission excited at 310 nm were: retinol (λ = 490 nm, FWHM = 112 nm), arachidonic acid (λ = 425 nm, FWHM = 120 nm), oleic acid (λ =370 nm, FWHM = 85 nm), linoleic acid (λ = 417 nm, FWHM = 92 nm), and for excitation at 366 nm: retinol (λ = 490 nm, FWHM = 112 nm), arachidonic acid (λ = 470, nm, FWHM = 93 nm), oleic acid (λ = 462 nm, FWHM = 90 nm), and linoleic acid (λ = 428 nm, 73 nm) [25]. These bands were combined with an additional GMG function describing the emission at wavelengths shorter than 420 nm. This spectral component is generally ascribed to proteins and made free to adapt its profile to achieve the goodness of fitting. The achievement of the satisfying combination of GMG functions required the addition of minor curves. One of these, with a maximum peak value around 440 nm, was difficult to ascribe to a defined fluorophore despite falling in the emission range of fluorescing fatty acids or NAD(P)H and was thus indicated as undefined. A band peaking at around 560 nm was detected only in I/R serum samples excited at 366 nm, possibly ascribable to flavins released by injured hepatocytes or to bilirubin [43,49,57,58]. In addition, a negative band with a peak of around 410 nm was likely due to light absorption by minor hemoglobin, as a result of the not-completely-avoidable hemolysis occurring during the blood processing to obtain serum. The minor added curves covered a spectral area lower than 10% or 3%, for 310 nm or 366 nm excitation, respectively, as compared with the overall processed spectral area.

#### 4.3.3. Serum AF Spectrofluorometric Analysis for Bilirubin

The AF excitation spectra from the crude serum were collected at 570 nm, the wavelength already assessed to favor the detection of bilirubin spectral changes [43,52]. The curve-fitting analysis procedure had been performed according to the GMG functions already defined from the analysis of bilirubin in solution with solubilizing agents and albumin [43]. A dominating band (λ = 330 nm, = 60 nm), likely ascribable to mixed longer excitation tails of fluorescing fatty acids and retinol, was considered along with the three minor nm bands ascribable to bilirubin AF (λ = 400 nm, FWHM = 35 nm; λ = 435 nm, FWHM = 40 nm; λ = 470 nm, FWHM = 70 nm). 

### 4.4. Serum Fatty Acid Quantification by GC/MS 

The fatty acid profile has been analyzed by a DSQII GC/MS system (TraceDSQII mass spectrometer, TraceGCUltra gascromatograph, ThermoFisher Scientific, Waltham, MA, USA), the Xcalibur MS Software Version 2.1, including NIST Mass Spectral Library (NIST 08), and the Wiley Registry of Mass Spectral Data 8th Edition for the assignment of chemical structures to chromatographic peaks. To proceed to the fatty acid assay, 50 µL aliquots of serum were dissolved in 1 mL methanolic HCl (2N) in reaction vials. The vials were capped and heated at 70–80 °C for 4 h. The samples were allowed to cool, then they were dried under a nitrogen stream. Later, 250 µL of dichloromethane was added and 1 µL aliquot was sampled for analysis. Dichloromethane alone was used as a blank to avoid carryover from the previous analysis. The reference standard Marine Oil FAME Mix from Restek S.r.l. 20063 Cernusco sul Naviglio MI, Italy (cat. 35066) was used to identify and quantify the fatty acids. The multianalyte standard solution was 10–160 µg/mL in hexane. Each identified peak was expressed as a relative percentage area of total methylated fatty acids (FAME).

### 4.5. Serum Bilirubin Biochemical Assay

The bilirubin levels in the serum were determined by means of a bilirubin total assay kit, according to the supplier’s instruction (Diazyme Europe, GmbH, Dresden, Deutschland). 

### 4.6. Statistical Analysis

Unistat (Unistat^®^ Statistical Package, Version 6.5 04, Unistat Ltd., London, England) and R software (R Development Core Team) were used to perform the statistical analysis. The value of *p* < 0.05 was considered statistically significant. 

## 5. Conclusions

The changes in autofluorescence signals relatable to the fluorescing fatty acids, retinoids, and bilirubin detected in a crude serum obtained from rats submitted to hepatic I/R are reversed by the OCA treatment. These findings are in agreement with the protective and anti-inflammatory activities of this potent FXR agonist and they allow us to propose that fluorescence has an ability to monitor the changes in fluorescing fatty acids, arachidonic acid in particular, as a supportive tool of pharmacological studies in hepatology, although additional studies should be performed with other substances and in other models of liver diseases.

## Data Availability

Data supporting reported results are stored in magnetic memory mass-, the data presented in this study are available on request from the corresponding author.

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
