# Peer review of "Fatty Acids and Bilirubin as Intrinsic Autofluorescence Serum Biomarkers of Drug Action in a Rat Model of Liver Ischemia and Reperfusion"

_molecules, 2023, doi:10.3390/molecules28093818_

Round 1

Reviewer 1 Report

This article is devoted to the study of autoflorescence of fatty acids and bilirubin in the serum when exposed to obeticholic acid in the liver ischemia/reperfusion model in rats. The authors propose to use the ability of substances to autoflorescence as an aid in evaluating the effects of pharmacological treatment in serum in real time.

After a careful study of the data presented, I believe that in this version the article cannot be accepted for publication.

The Reasons:

1.      What is the purpose of this auxiliary test if there are methods for determining the concentration/presence of specific compounds in the serum for certain pathologies/treatments?

2.     It seems to me that there is not enough data to declare a new diagnostic method:

Only one substance and its effect only on liver ischemia/reperfusion in rats has been considered. Would there be similar changes with other substances and in other models of pathology? It would be great if the authors could make a comparison between the observed serum autofluorescence and fatty acid concentrations, retinol, (as was done for bilirubin in Figure 5), for example, using high performance chromatography-mass spectrometry. (show the relationship between autofluorescence and the concentration of these substances).

3. Figures 1 and 3 partially duplicate the data in your article doi.org/10.3390/molecules25061327 (Fig. 4 and Table 3), except for the addition of data from obeticholic acid. The article "Spectrofluorometric Analysis of Autofluorescing Components of Crude Serum from a Rat Liver Model of Ischemia and Reperfusion" is more complete and complete compared to this material where autofluorescence of pure solutions of fatty acids and retinol as well as these same substances in serum are given.

4.      Line124-125. while the spectral component ascribable to proteins was identified under 310 nm excitation, and linoleic acid under 366 nm excitation (Figure 2A, 2B). Where exactly is the spectral component attributed to proteins shown?

5.   The data in fig. 4,5 are published in part in doi.org/10.1016/j.jphotobiol.2020.112121 (Fig. 4 A-B; Table 2).

6.       What is the contribution of albumin to bilirubin autoflorescence in animal serum in normal and pathological conditions?

7.      Statistical analysis: * p ≤ 0.05 versus sham operated; versus OCA IR 60/60 min; ** versus OCA IR 60/60 min. * cannot denote sham operated and compared to the sham operated, etc.  The resolution of the figure captions is poor and the number of experiments (n) is not listed everywhere.

Author Response

Reviewer 1

This article is devoted to the study of autoflorescence of fatty acids and bilirubin in the serum when exposed to obeticholic acid in the liver ischemia/reperfusion model in rats. The authors propose to use the ability of substances to autoflorescence as an aid in evaluating the effects of pharmacological treatment in serum in real time.

After a careful study of the data presented, I believe that in this version the article cannot be accepted for publication.

We sincerely thank the Reviewer for the remarks to our manuscript, helping us to improve our paper. 

We have duly considered all the questions, and revised the text accordingly. Here below please find our point-by-point answers and indications about changes made to the text, highlighted in yellow.

The Reasons:

Q 1.      What is the purpose of this auxiliary test if there are methods for determining the concentration/presence of specific compounds in the serum for certain pathologies/treatments?

A 1. The purposes and advantages of AF based measurements are given in the revised “Abstract” (lines 33-34), and made it clearer in the revised “Introduction” text (lines 92-97). 

  1. It seems to me that there is not enough data to declare a new diagnostic method:

Only one substance and its effect only on liver ischemia/reperfusion in rats has been considered. Would there be similar changes with other substances and in other models of pathology? It would be great if the authors could make a comparison between the observed serum autofluorescence and fatty acid concentrations, retinol, (as was done for bilirubin in Figure 5), for example, using high performance chromatography-mass spectrometry. (show the relationship between autofluorescence and the concentration of these substances).

A.2 We thank the reviewer for the remark about “new diagnostic method”, helping us to improve the text at the end of “Conclusions” on the need of further validation with “additional studies should be performed with other substances and in other models of liver diseases.” (lines 424-425).

According to the suggestion on additional analysis of fatty acid concentrations, GC/MS has been performed for the quantification of arachidonic acid, linoleic acid and oleic acid and the results reported in the new figure 4 with the related text (lines 170-184). Since the serum samples were available only for one kind of analysis (fatty acids by GC/MS or retinol by HPLC), we preferred to search a support to the AF data on fatty acids, considering their importance as precursors of signaling agents of inflammation and damage.

Q 3. Figures 1 and 3 partially duplicate the data in your article doi.org/10.3390/molecules25061327 (Fig. 4 and Table 3), except for the addition of data from obeticholic acid. The article "Spectrofluorometric Analysis of Autofluorescing Components of Crude Serum from a Rat Liver Model of Ischemia and Reperfusion" is more complete and complete compared to this material where autofluorescence of pure solutions of fatty acids and retinol as well as these same substances in serum are given.

A 3. - We agree with the Reviewer on this remark. The spectra in Figure 1A,B are examples taken from the pool of spectra measured for each group of treatment, and this is now stated in the legend. Also, spectra from the OCA treated group were never reported in other manuscript. Also, a notice on the former results on AF of serum was already given in the introduction (lines 92-95), and indicated to offer the basis with interesting insights for the development of the real-time, cost-effective serum AF analysis as a diagnostic supportive mean in assessing the effects of pharmacological treatments. The related text has been revised to better stress this concept.

Q 4.      Line124-125. while the spectral component ascribable to proteins was identified under 310 nm excitation, and linoleic acid under 366 nm excitation (Figure 2A, 2B). Where exactly is the spectral component attributed to proteins shown?

A 4. – We agree with the Reviewer that former description for the AF detection of proteins, namely albumin, was unclear. The related sentences have been revised and rephrased:  lines 138-141; lines 338-340). A new reference (Ref 56) has also been added.

Q 5.   The data in fig. 4,5 are published in part in doi.org/10.1016/j.jphotobiol.2020.112121 (Fig. 4 A-B; Table 2).

A 5.  We agree with the Reviewer on this remark. The spectra in Figure 4A are examples taken from the pool of spectra measured for each group of treatment, and this is now stated in the legend. In Figure 4B, an example of fitting analysis is shown for a spectrum from the OCA treated group, never reported in other manuscript.

Q 6.       What is the contribution of albumin to bilirubin autoflorescence in animal serum in normal and pathological conditions?

A 6. The autofluorescence of bilirubin is well assessed to be greatly enhanced upon binding to albumin, and the different excitation / emission properties of the two fluorophores allow to exclude the detection of a contribution of albumin fluorescence to that of bilirubin (for example see Ref. 49, Plavskii, et al., doi:10.1007/s10812-007-0019-6.s; Ref. 66, Athar et al., 1999; Tayyab et al., 2003, at: “DOI 10.1016/S0141-8130(02)00081-8, showing that no fluorescence is given by albumin when excitation for bilirubin is applied”, not cited in our present proposed paper).

As to pathological conditions, an increase in blood bilirubin is found in new born jaundice and more generally in cholestasis induced by various causes. These events can result in an imbalance between albumin and bilirubin. Amin et al., 2011, for example, reports on technologies to measure bilirubin in new born jaundice, including a method using a Hematofluorometer based on the AF of bilirubin binding to albumin (DOI: 10.1053/j.semperi.2011.02.007; Ref 44 in our present proposed paper).

Q 7.      Statistical analysis: * p ≤ 0.05 versus sham operated; versus OCA IR 60/60 min; ** versus OCA IR 60/60 min. * cannot denote sham operated and compared to the sham operated, etc.  The resolution of the figure captions is poor and the number of experiments (n) is not listed everywhere.

A 7. The resolution of Figure captions has been improved. The number of experiments is now given in the added scheme 1. We apologize for the lack of clarity for statistical analysis: the related sentences have been revised.

Reviewer 2 Report

Croce et al., presented a fluorescence spectra-based method to assess the diagnostic potential of the combined AF analysis of fluorescing fatty acids and bilirubin performed in the serum to estimate their changes as reference parameters of OCA-induced pharmacological effect in a rat model of liver ischemia and reperfusion. The reviewer has several comments before its consideration of publication.

Figures 2 and 5: Could the authors include sample size in the captions?

Figures 2 and 3 are identical. Please double-check Figure 3 to make sure it’s the correct figure.

Line 172: Figure 4A should be corrected to Figure 4B

Line 250: Figure 4C should be corrected to Figure 5C

Well written.

Author Response

Reviewer 2

Croce et al., presented a fluorescence spectra-based method to assess the diagnostic potential of the combined AF analysis of fluorescing fatty acids and bilirubin performed in the serum to estimate their changes as reference parameters of OCA-induced pharmacological effect in a rat model of liver ischemia and reperfusion. The reviewer has several comments before its consideration of publication.

We sincerely thank the Reviewer for the remarks to our manuscript, helping us to improve our paper. 

We have duly considered all the questions, and revised the text accordingly. Here below please find our point-by-point answers and indications about changes made to the text, highlighted in yellow.

Q- Figures 2 and 5: Could the authors include sample size in the captions?

A – Sample size are now given in the added scheme 1 summarizing the experimental plan

Q - Figures 2 and 3 are identical. Please double-check Figure 3 to make sure it’s the correct figure.

A – Sorry for the mistake and many thanks for signaling. The Figure 2 is now correct.

Q - Line 172: Figure 4A should be corrected to Figure 4B.

A – Sorry for the mistake and many thanks for signaling. The text has been corrected.

Q - Line 250: Figure 4C should be corrected to Figure 5C

A – Sorry for the mistake and many thanks for signaling. The Figure 5 has been reorganized, as well as the related legend, and the related text has been corrected.

Reviewer 3 Report

Review report

Please find in the following my comments about the review of a manuscript under the title (Fatty acids and bilirubin as intrinsic autofluorescence serum biomarkers of drug action in a rat model of liver ischemia and reperfusion). In this study, the authors aim to investigate the diagnostic potential of the combined AF analysis of fluorescing fatty acids and bilirubin performed in the serum to estimate their changes as reference parameters of OCA-induced pharmacological effect in a rat model of liver ischemia and reperfusion.

Originality and relevance

§  The study is interesting for reading.

§  The study has moderate scientific quality.

§  The study is relevant to the scope of this journal.

§  The manuscript is relevant to the field and its presentation needs some modifications to be clearer.

Some points in the study need to be added and changed to adhere to the journal's standards. If the authors will revise with proper objectives, organization and supportive evidence, hypothesis, and their claim clearly in abstract, introduction, results, discussion, and conclusion, it may be considered for publication

Comments:

Abstract:

§  Abstract is not clear. Add more details about the methodology and mentioned the most prominent finding in the results.

Introduction

§  The rationale of the study is not clear in the introduction section.

Materials and methods

§  Add references for all methods used.

§  It is better to draw a scheme illustrating the experiment groups, doses, and duration.

Results:

§  The results section needs more illustration of the findings.

§  The title and legends of the figures should be informative and self-explanatory? Revise.

Discussion:

§  Discussion needs more interpretation of the results.

§  Add a section under the title (Limitation of study).

§  Add your recommendation at the end of the conclusion section.

§  Add a list of abbreviations.

Moderate editing of English language and style required. There are some typos and grammatical errors. The punctuation should also be checked

Author Response

Reviewer 3

Please find in the following my comments about the review of a manuscript under the title (Fatty acids and bilirubin as intrinsic autofluorescence serum biomarkers of drug action in a rat model of liver ischemia and reperfusion). In this study, the authors aim to investigate the diagnostic potential of the combined AF analysis of fluorescing fatty acids and bilirubin performed in the serum to estimate their changes as reference parameters of OCA-induced pharmacological effect in a rat model of liver ischemia and reperfusion.

We sincerely thank the Reviewer for the remarks to our manuscript, helping us to improve our paper. 

We have duly considered all the questions, and revised the text accordingly. Here below please find our point-by-point answers and indications about changes made to the text, highlighted in yellow.

 Originality and relevance

  • The study is interesting for reading.

  • The study has moderate scientific quality.

  • The study is relevant to the scope of this journal.

  • The manuscript is relevant to the field and its presentation needs some modifications to be clearer.

Some points in the study need to be added and changed to adhere to the journal's standards. If the authors will revise with proper objectives, organization and supportive evidence, hypothesis, and their claim clearly in abstract, introduction, results, discussion, and conclusion, it may be considered for publication

Comments:

 Abstract:

  • Abstract is not clear. Add more details about the methodology and mentioned the most prominent finding in the results.

A – The abstract has been revised.

 Introduction

  • The rationale of the study is not clear in the introduction section.

A – The sentences at the bottom of this section have been implemented (lines 92-107).

Materials and methods

  • Add references for all methods used.

A – Details for bilirubin assay have been added – lines 271-272

  • It is better to draw a scheme illustrating the experiment groups, doses, and duration.

A - A scheme (Scheme 1) has been added to the text to summarize the experimental plan.

Results:

  • The results section needs more illustration of the findings.

A – Parts of the text have been revised and implemented to better illustrate the findings. Implemented text is highlighted in yellow.

  • The title and legends of the figures should be informative and self-explanatory? Revise.

A – Legends have been implemented.

Discussion:

  • Discussion needs more interpretation of the results.

A – Discussion has been Implemented, text highlighted in yellow.

  • Add a section under the title (Limitation of study).

A – According to the suggestion we stressed the possible limitation of the AF based studies by implementing the text (226-234).

  • Add your recommendation at the end of the conclusion section.

A - As suggested we modified the conclusion, adding that to further support our results additional studies should have to be performed with other substances and in other models of liver diseases.

  • Add a list of abbreviations.

A – The list of abbreviations has been added

Round 2

Reviewer 1 Report

Dear authors,

thank you for your responses to the comments. Undoubtedly your corrections have improved the article. There are still typos in the article, e.g., two figures 4.

Author Response

 Q- Thank you for your responses to the comments. Undoubtedly your corrections have improved the article. There are still typos in the article, e.g., two figures 4.

A- Many thanks for the work and time spent on our paper, we feel surely helped us to improve it.

The second "(Figure 4)" has been removed. The text has been also checked for other typos

Reviewer 2 Report

The authors have fully addressed my concerns. I have no further comment.

Author Response

Many thanks for the work and time spent on our paper, we feel surely helped us to improve it.

Reviewer 3 Report

Please find in the following my comments about the review of a manuscript under the title (Fatty acids and bilirubin as intrinsic autofluorescence serum biomarkers of drug action in a rat model of liver ischemia and reperfusion). 

The authors revised their manuscript. I think the revisions have been completely conducted according to the reviewers' comments. There are no more comments required.

Minor editing of English language required

Author Response

(The authors gave the same response as above.)
